# Quality and Safety of Proximity Care Centered on the Person and Their Domiciliation: Systematic Literature Review

**DOI:** 10.3390/ijerph20247189

**Published:** 2023-12-17

**Authors:** Carlos Martins, Ana Escoval, Manuel Lopes, Susana Mendonça, César Fonseca

**Affiliations:** 1Cova da Beira Group of Health Centers, 6200-034 Covilhã, Portugal; 2Comprehensive Health Research Centre, University of Évora (CHRC-UE), 7000-811 Évora, Portugal; anaescoval@ensp.unl.pt (A.E.); mjl@uevora.pt (M.L.); 3Department of Nursing, University of Évora, 7000-811 Évora, Portugal; susana.mendonca@uevora.pt (S.M.); cfonseca@uevora.pt (C.F.)

**Keywords:** quality, safety, home health care, patient safety, home nursing

## Abstract

The quality and safety of health care are a priority for health organizations and social institutions to progressively provide people with a higher level of health and well-being. It is in the development of this path that home care currently represents an area of gradual investment and where health care services and the scientific community have shown interest in building circuits and instruments that can respond to needs. The purpose of this article is to identify areas and criteria for quality and safety in home care. The method used was a systematic review registered in PROSPERO (CRD42022380989). The search was systematically carried out in CINAHL Plus with Full Text, MEDLINE with Full Text and Psychology and Behavioral Sciences Collection, using the following criteria: articles published in Portuguese and English, from January 2017 to November 2022. The results of the analysis of the articles showed areas of quality and safety in home care with their respective dimensions and operational criteria. We concluded that there are three areas: the intervention with the patient, with proximity and patient-centered care, which integrates the individual care plan and the proximity of professionals to the patient and family; the intervention of care and service management, with care management and clinical governance that includes the integrated model of health care, goal management, and context management; and the intervention related to training and professional development, where we have the skills and training of professionals.

## 1. Introduction

In recent years, shifts in demographic patterns and the prevalence of chronic diseases have led to new “health needs and an increase in the complexity of health problems, associated, among others, with population aging and multiple morbidity and dependence, but also to a more acute awareness of access to health as a right” [1]. These nuances and complexities gradually drive healthcare towards a people-centered approach, one that respects and addresses individual preferences, needs and values [2]; this entails providing individuals with access to the type and intensity of care they genuinely require, in the right location and at the appropriate time and underscores the importance of continuity of care, ensuring that care is suitable for transitions between different levels of healthcare [3].

For these reasons, home care currently represents a progressive investment area where healthcare services and the scientific community express concern and interest in developing knowledge that enables the creation of circuits and tools to ensure the quality of care [4,5]. Additionally, there is a focus on building strategies to prevent adverse events in home care [6].

Home care is “defined as care that aims to meet the social and health needs of people in their own homes” [1]. These can respond to ongoing challenges arising from changes in epidemiological and demographic profiles, prompting the need to reflect and rethink local health intervention strategies in constant paradigm shifts at the scientific, technical and organizational levels, centered on the person and his context [7]. From the perspective of care at home, where the health professional travels to conduct interventions and engage with the patient and family throughout the care process, as a way of guaranteeing care in the absence of professionals. These circumstances contribute to the increased complexity of the team’s work, as it is necessary to train the patient and family for the continuity of care. There is also a change in the context of care, which becomes the home of each patient, leading us to a critical and reflective analysis of the care process due to the specificities that can bring about a change in the context where care takes place. Well, each patient’s home is unique. The interaction among different actors occurs in people’s homes, where the patient and the family caregiver are not only recipients of care but also integral partners in the care team.

To respond to this complexity, health services must adapt and rethink their intervention in collaboration with different institutions and community resources, adopting a transdisciplinary perspective. Establishing dialogue between different areas of knowledge and understanding processes in a new attitude to understand and respond to the multidimensionality of human beings and the world. This involves fostering dialogue between different areas of knowledge and understanding processes, encouraging a pluralistic approach to knowledge for a broader exercise of human cognition [8].

It is also important to maintain the connection between patients who need to care at home and their families/caregivers, with professional care providers. This aspect has been a significant concern, and at the same time presents itself as a level with potential for development. Gaining, in recent times, greater relevance with the eHalth, which focuses on the impact of information and communication technologies in the field of health [9].

As Alonso et al. [9] refer to, the immense availability of information systems and the wide use of technologies by patients, family and/or caregivers have triggered the need to develop literacy around technology and information systems. The vast availability of medical information systems has facilitated communication between all actors in the care process. The height of information and communication technologies and the computerization of health services, where people’s records are digitally stored securely in the so-called Electronic Health Record, thus allowing multiple authorized users to simplify the management of information and its transfer between the different services and professionals. At the same time, we have mobile applications and Ambient Assisted Living assistive technology, which promote the empowerment of people’s capabilities through digital environments and could also have a significant impact on the provision of home care [9].

At this juncture, patients may be exposed to wide variations in the quality of care they receive. To circumvent these variations, it is essential to gather evidence that determines the effectiveness of quality, costs and risks. The lack of a scientific approach can lead to results that are opposite to those intended by improvement initiatives [10].

This challenges the traditional trilogy of quality and safety—structure, process and results [11]—necessitating a reevaluation because ensuring the quality and safety of home care is unquestionably crucial.

In this context, the Donabedian Quality Model [11] integrates structure, process and result indicators into systems theory, adapting them to the health sector. The structure pertains to how the organization presents itself concerning resources, norms, routines and the system of values and expectations essential for the care process. The process is related to the way health care is provided to the patient, following established technical-scientific standards. The result corresponds to the consequences of activities carried out at home or by the professionals involved [12,13,14].

The perspective of American Institute of Medicine, in 1990, defines quality as “the extent to which health services provided to individuals and populations increase the probability of obtaining the desired health outcomes, and are consistent with current professional knowledge” [15], extending this concept to the general population and to the consistency of results in the organization.

Regarding the evaluation of the quality of care, the World Health Organization [15] defines more consensual dimensions based on the American Institute of Medicine: Efficiency, Effectiveness, Equity, Focus on the person, Safety and on time Right/Opportunity.

In Portugal, existing models of quality and safety in the care delivery process have been created and approved for areas of health care other than home care [1]. Also, the review carried out by Al Anazi et al. shows a clear gap in the literature on the quality of home care in Arab countries, emphasizing the need for more studies, especially quality studies on care in home health environments [16].

Therefore, it becomes imperative to develop a care quality model that considers the great variability in terms of structure and actors in the care process—a relational model in a context of care that is always different.

Researching the international literature on quality and safety in home care, some authors highlight two dimensions. First, the perspectives of the sick person and caregivers regarding home care, as shown in studies carried out by LaFave et al., Sanerma et al., Bolenius et al. in Sweden, Róin in the Faroe Islands, Dostálová et al. in the Czech Republic, and Oosterveld-Vlug et al. [17,18,19,20,21,22]. Second, the perspective of professionals on quality and safety in home care is evident in studies conducted by Olsen in Norway and by Al Anazi et al. in Arab countries [16,23].

These results also show more specific aspects in relation to home care, such as the importance of knowing and understanding the patient’s path, the need to understand the role of home care providers, the importance of implementing quality checklists, and the need for further studies in this area of home care [16,23].

In terms of consensus, according to the World Health Organization [15], quality health care can be defined in different ways, but there is a convergence on the recognition of quality health services around the world, which must be: effective, providing care based on evidence; insurance, in order to avoid harming the person being cared for; centered on people, with the intention that care responds to their needs, values and preferences. Furthermore, for the benefits of quality health care to be real, health services in general, and similarly, home care, will have to be: opportune, in order to reduce waiting times and, especially, harmful delays; equitable, providing quality care regardless of sex, gender, age, ethnicity, race, language, religion, political option or geographic location; integrated, mainly providing care centered on the overall recovery of the person throughout life, sequentially at all levels of health care; and efficient, in order to maximize the benefit of available resources and reduce waste.

The World Health Organization [15] also mentions that since there are many definitions of quality, these may have a broad enough interpretation to define the perception of quality at the national, regional or health unit level. Noting that each country may have its own local understanding or definition of quality, its quality policy and a corresponding strategy that responds to local needs [15].

In summary, the concept of quality over time has taken on different definitions and perspectives. In other words, issues related to quality in health and safety have mobilized growing concern on the part of policymakers, professionals and patients and their families.

This systematic review aims to identify the areas and criteria that may be involved in the quality and safety of care provided at home, with the aim of contributing to scientific knowledge on the subject, identifying guidelines for evidence-based practice and informing possible lines of research. Future, and namely, to contribute to the construction of a quality and safety model for home care in primary health care. To achieve this, the review addresses the following questions:-What is the state of the art in the areas of quality and safety in home care?-What areas and criteria should exist for the construction of a health care model that guarantees the quality and safety of patients at home?

## 2. Materials and Methods

### 2.1. Protocol and Registration 

The protocol for this review was registered in the International Prospective Register of Systematic Reviews (PROSPERO) (CRD42022380989), and an article was subsequently prepared and has already been published.

### 2.2. Study Design

This systematic review was developed in accordance with Preferred Reporting Items for Reporting of Systematic Reviews and Meta-Analyses (PRISMA) [24] and the methodology followed this reference.

Considering that the scope of this study is wide, we chose to include in this review all types of primary quantitative or qualitative empirical studies, including cross-sectional, longitudinal, observational or experimental studies. This review also included studies with and without a comparison group.

### 2.3. Eligibility Criteria

Inclusion criteria were all studies published from January 2017 to November 2022 trying to be as current as possible, written in Portuguese and English, as we were looking for similarity in care systems, and it seemed to us that they would be covered in articles written in Portuguese and English, including studies on areas and criteria of quality and safety in home care, in different geographic areas, communities, cultures or specific environments, with different methodologies of work and organization of health and social services. The main aim is to identify areas and criteria for the quality and safety of home care and to subsequently identify the most appropriate care in a home context.

Exclusion criteria were all studied that were not directly related to the research aim or research questions, non-primary studies, all those that did not present an abstract or complete article, and those whose study design did not meet the defined criteria.

### 2.4. Search Strategy

As a strategy, it was intended to carry out a wide bibliographical research and database consultation: CINAHL Plus with Full Text, MEDLINE with Full Text and Psychology and Behavioral Sciences Collection.

### 2.5. Search Terms and Boolean Operators

The search was carried out in the CINAHL Plus with Full Text, MEDLINE with Full Text and Psychology and Behavioral Sciences Collection databases, according to the Medical Subject Headings (MeSH) terms: Home care, quality or safety. Subsequently, the search strategy included the following MesH descriptors or similar terms: “Home care services”, “Housing”, “Resistant homes”, “Home nursing”, “Home care”, “Patient safety”, “Indicators of patient safety”, “Safety, “Quality of service”, “Quality health”, “Quality indicators”, and “Quality”.

The research combined the key concepts of the investigation questions with the terms: (“Home care services”) OR (“Housing”) OR (“Resistant homes”) OR (“Home nursing”) OR (“Home care”) AND (“Patient safety”) OR (“Patient safety indicators”) OR (“Safety”) AND (“Quality of service”) OR (“Quality healthcare”) OR (“Quality Indicators”) OR (“Quality”) AND (“IT Quality”).

### 2.6. Data Collection and Analysis

#### 2.6.1. Selection of Studies

All non-primary studies and those that were not directly related to the research aim or research questions, those that did not present an abstract or complete article and those whose study design did not meet the defined criteria were excluded.

Study selection included several steps. The studies resulting from the search in each database were exported to Mendeley, and duplicates were removed using an electronic methodology in addition to the manual activity of two reviewers by reading the articles. Then, two reviewers independently assessed the inclusion of studies by reading the titles, abstracts and keywords, excluding those that did not meet the inclusion criteria (Figure 1—PRISMA Flowchart) [24]. Since there was consensus on the selected articles, it was not necessary to resort to a third reviewer. Subsequently, the full text evaluation phase was carried out using the same method; in order to minimize biases, it was also not necessary to use a third reviewer.

#### 2.6.2. Data Extraction

Similarly, data extraction was performed by two of the reviewers responsible for selecting the studies independently, with no need for a third reviewer due to the lack of disagreements. In this phase, a descriptive evaluation of each study was carried out using an extraction tool built to extract pertinent information according to the research questions.

#### 2.6.3. Quality Assessment

We used the evaluation tools of the Joanna Bring Institute (JBI) [25] to analyze the quality of the articles included in this review, which were performed by the two reviewers independently, with no disagreements. The choice of these specific tools was due to the different methodologies of the studies included, for allowing us to evaluate the methodological quality of each study and determine the extent to which the study addressed the possibility of bias in its design, conduct and analysis. JBI’s approach considers the best available evidence, the context in which care is delivered, the individual patient and the professional judgment and expertise of the health professional.

#### 2.6.4. Data Synthesis Strategy

The studies had different methodologies; the synthesis of data and analysis of the results was of a narrative nature, structured to answer the defined questions, allowing an individual analysis of the content and/or structure of the studies until synthesizing a metanarrative [26]. A quantitative analysis was not feasible.

## 3. Results

The search resulted in 183 articles, and after removing 44 duplicates, it resulted in 139 publications. Studies were excluded for the following reasons: 1st the title was not directly related to the research, aim or research question *(n* = 105); 2nd being a non-primary study (*n* = 8); after that, 26 articles resulted, which were submitted to full reassessment and reappraisal, thus obtaining, for eligibility, 11 articles (*n* = 11), which were read in full, where they were 5 studies were excluded (*n* = 5) because they did not respond to the review questions and did not have conclusions in relation to the areas and criteria of quality and safety in home care, thus being the 3rd reason for exclusion. In this sense, 6 studies were selected (*n* = 6) [27,28,29,30,31,32] that met the inclusion criteria (Figure 1—PRISMA 2020 flow diagram).

Of the six articles selected and included in the review, two were from the MEDLINE database [27,29] and four were from CINAHL Plus [28,30,31,32], all in English, from different countries, Canada (*n* = 1), Holland (*n* = 1), Italy (*n* = 1), England (*n* = 1) and Norway (*n* = 2). In chronological terms, by year of edition or last revision of the articles, two corresponded to 2022; two to 2020; one to 2021; and one to 2019. All focused on the quality and safety of patient care at home, with two articles emphasizing patients aged 65 or over.

The systematic review included an analysis of six articles on the quality and safety of home care, which resulted in the findings shown in Table 1.

The results of the articles were submitted to analysis, from which three areas emerged, as well as their dimensions and criteria to ensure the quality and safety of care for patients in a home situation. In the first area, we have Intervention with the Patient, with a dimension related to proximity and patient-centered care, integrating two criteria: the individual care plan and the proximity of professionals to the patient and family. In the second area, Management of Care and Services, with the dimension of care management and clinical governance of services, which integrates three criteria: integrated health care model, goal management and context management. In the third area, Training and Professional Development included one dimension, the skills and training of professionals.

Next, we present Table 2 in more detail with the areas found, the dimensions and their operationalization criteria.

In addition to the findings in relation to the research questions, the articles also showed some of the obstacles that arise when implementing processes to guarantee the quality and safety of patient care at home, which we present in Table 3.

## 4. Discussion

The results of this systematic review suggest that there are areas, dimensions and criteria that, if put into practice, allow professionals to guarantee the quality and safety of primary health care in the intervention of people living at home. Thus, we have three areas of quality and safety that integrate the dimensions and multiple criteria that we present.

First Area—Intervention with the Patient.

In the area of Intervention with the Patient, within the proximity and patient-centered care dimension, two criteria stand out: the individual care plan and the proximity of professionals to the patient and family. This dimension holds extraordinary value in ensuring quality and safety across various contexts of care, including in-home care. In this context, the professional, to guarantee quality and safety, dedicates himself exclusively to that patient, to his problem, and is imbued with understanding his needs and wishes. This relational process that develops between the patient and the health professional is very close and is facilitated by taking place in the patient’s home and within the family. This proximity fosters a robust and trusting relationship, potentially enhancing adherence to care and, consequently, elevating overall quality, safety and satisfaction. 

All the aspects mentioned reflect what we found in the operationalization criteria of this first area: the importance of the individualized care plan in home care, where care is individualized [28], where there must be continuity and organization of work [27], in which interpersonal aspects are crucial [32], as well as the assessment of multidimensional needs [28,30]. These individualities are structuring pillars for obtaining quality and safe home care, in which the individualized care plan is a fundamental instrument for the different reasons presented and also for being able to facilitate the integration and management of the patient’s journey in the different types of health care. Individualized care plans are also particularly useful for people with multiple health conditions [1]. At the same time, we say that the individualized care plan allows for individual planning of the care process and effective communication between all the different levels of care and the caregivers involved. Therefore, in this care, outside health institutions, it assumes high importance. On the other hand, at home it is essential that the planning of interventions is comprehensive and dynamic, considering the evolution of the situation of the patient, family or caregiver, Brunelli et al., Kattouw and Wiig, and Malley et al. [28,30,32]. Home care should be based on a holistic approach to all problems, centered on the patient and allowing an understanding of the complexity of their problems and needs [33]. Isolated interventions should be avoided, as they are sometimes ineffective because they are not appropriate to the patient’s global needs and inefficient due to the inadequate use of resources, which affects the quality of the service and care provided.

The individualization and centrality of care on the patient promotes better adequacy in the management of the path and greater effectiveness in the integration in the health system, as it is more adequate, and in tune, with what is specifically needed. On the other hand, it is important to have a vision of the effectiveness of the care provided through strategies for monitoring and evaluating home care [28]. In addition to evaluating, it is also essential to build bridges of continuity and organization of care [27] so that the transition processes experienced by patients are as painless as possible. As highlighted by the World Health Organization [12], integrated care must be coordinated at all levels and by all providers to be safe, limiting lapses and adverse events as much as possible.

At the same time, in home care, the proximity of professionals to the patient and family [30] contributes to the quality and safety of health care at home, absolutely requiring the involvement of the patient and family in the care process since monitoring and warning signs are triggered by the patient and/or family. In other words, in the absence of the professional, they assume the continuity of care and their partnership is essential to limit risks in the patient’s clinical situation. In fact, health professionals often only go home to carry out scheduled interventions or respond to warning signs. Therefore, it is of greater importance to train the patient and family to act in the absence of a health professional. Naturally, this relational model between the various actors takes place in the context of people’s homes, which requires the integration of the patient and family caregiver as team partners and simultaneously as care recipients, which facilitates the establishment of a greater relationship of proximity and trust [30].

On the other hand, care at home and in proximity, according to Brunelli’s study et al. [28] present themselves as potential facilitators of patient and family access to the health system and to more equitable care. On the other hand, it is extremely important to trigger evaluation mechanisms, namely more narrative evaluation methods that support open communication about the care experience [29] and their perceptions about home care, which are associated with quality of care outcomes [34]. In summary, the centrality of patient care and the proximity of health professionals to the patient and his family or caregiver are vital to guarantee the quality and safety of the service and home care, where a relationship of proximity and trust is privileged, in which assessment instruments can be of enormous help in improving intervention strategies and mechanisms.

Second Area—Management of Care and Services.

In the area of Care Management and Clinical Governance of Services, we integrate three criteria: the integrated health care model; the management of goals; context management.

The importance of creating a care model is highlighted by Brunelli et al. [28] who proposed that an integrated tool for accreditation in health and social assistance would help to improve the quality of home care, making the quality of life of patients better and safer, with an agreement of 100% of the heard experts. From this perspective, we believe that creating an integrated model of health care and social care is fundamental, as it helps to identify clearly define the goals to be met and defines guidelines for the type of management in the context where home care is provided. The review of the articles also showed that to ensure quality and safety of home care, it is essential to leverage the use of clinical data to improve the organization and manage resources and assess the goals to be achieved [31].

Certainly, still regarding the integrated home healthcare model, for it to function adequately and effectively, it is imperative that there is communication among professionals, patients, family/caregivers [27], and that information flows among the different stakeholders, as well as everyone collaborating in feeding the model around a common goal, which is the well-being of the patient. Additionally, the potential use of patient clinical data can limit lapses or failures in care regarding the multiple needs of the patient and facilitate the organization and management of the necessary human and material resources. In a study conducted by Silverglow et al. in Sweden, it became evident that the scarcity of information about the user between healthcare units and home care can constitute a barrier to the safety of healthcare [35].

To be more effective, the model should incorporate integrated care pathways capable of promptly addressing patients’ needs, mitigating prolonged waiting times, and averting unnecessary health risks. Additionally, it is imperative for health professionals to possess a comprehensive understanding of the community network and available resources [30], coupled with the expertise to seamlessly coordinate these elements to maximize the benefits of home care. Effective communication is paramount in the realm of home care, as highlighted by the study conducted by Silvesglow et al. demonstrated that communication problems, both between care professionals and with care unit managers, can weaken patient safety in home care [35]. In addition to these aspects, in the creation of an integrated model of healthcare at home, it is extremely important to invest in technology and artificial intelligence [31] to increase its effectiveness in the field, improve communication between the care provider and the patient/family/caregiver, to allow the professional, even from a distance, to monitor the patient’s evolution, and even promote greater safety in remote monitoring [31]. The greater the fidelity of the existing technology, the greater confidence the professional who is monitoring the patient from a distance can have. Simultaneously, the implementation of technology and artificial intelligence requires that there be a serious and broad investment in the training of professionals [29] and that the literacy of the patient and caregivers in this matter be increased. On the other hand, the use of technology can contribute to solving some problems in terms of human resource shortages and reducing errors. As mentioned by Kajander-Unkuri et al., using a robot for medication management had a notable effect on decreasing the use of working time by home care professionals [36].

Effective management and clinical governance of health services are essential to ensure quality and safety in home care [27,28,30,31], thus responding to current challenges arising from changes in epidemiological and demographic profiles. In this sense, it is pertinent to reflect and rethink local health intervention strategies in constant paradigmatic change at the scientific, technical and organizational level, centered on the person and their context [4].

Regarding the management of goals in home care, it is essential to act in accordance with the needs, considering the expected results to be achieved for that patient. The measures to be implemented to achieve these aims must focus on the quality and safety of care [29], as well as be built systematically for the organization of care [30]. It is also essential that monitoring and evaluation instruments are created in the management of goals [31] to have a vision and continuous evaluation of the expected and obtained results [30], which can be improved in the care process. These strategies make it possible to monitor and evaluate what is happening and, at the same time, to identify ways to improve the quality of care provided.

Context management in home care is of crucial importance. The context will always influence the way care is provided, better conditions promote better quality and satisfaction in care [27,30]. However, when the patient is institutionalized, the context is controlled and monitored by professionals specializing in the matter (beds, clothes, temperature, etc.), but at home, this does not happen, as we are in the patient’s natural environment, in which there are always conditions considered ideal for providing care. Therefore, it is essential that health professionals study the particularities of a person’s home and adapt it as much as possible to the patient’s needs [27], trying to minimize unnecessary interference as much as possible. Its main aim is to create favorable conditions for caregivers and families. It is important to always be aware of the value that the well-being of the patient and family assumes in their natural environment, therefore, we must interfere as little as possible. Additionally, leveraging community resources becomes essential in the context and exploitation of community resources [30].

Third Area—Training and Professional Development.

In the domain of Training and Professional Development, focusing on the dimension of skills and training of professionals, it is imperative to implement effective learning and training strategies that enhance professionals’ capabilities to meet the demands and nuances of the home care context [27,29]. professionals need to improve their background [29] in caring for patients and in the relationship with the family caregiver, who are both the focus of care and care partners. In fact, it is essential that the professionals who perform functions in home care are experts [29], who have knowledge of the most current protocols, who have decision-making capacity and who base their interventions on recent scientific evidence, thus preventing the fragmentation of their practice with unsatisfactory results. Therefore, the lack of a scientific approach can lead to results that are opposite to those intended [7], which can jeopardize patient safety. Consequently, there is a systematic need to enhance professional backgrounds [29], coupled with collaborative knowledge sharing [19].

Olsen’s study [19], exploring the perceptions and experience of health professionals regarding the quality and safety of home care, highlighted the importance of properly knowing the patient and understanding him as a whole in his journey, which is a key role for home care providers in the patient’s trajectory, that is, it is necessary to place what really matters to the patient in the care transition process [19]. Such achievement requires the existence of the necessary [27] and qualified resources. Therefore, it is natural that the adoption of a more scientific approach contributes to the improvement of quality and has great potential to promote the provision of high-safety care and to optimize the use of resources in health systems [7]. Therefore, to develop quality and safe home care, it is essential to invest in the training and development of professionals’ skills, optimizing the expertise of human resources [27] and having sufficient human resources at their disposal to guarantee quality home care to everyone who needs them. Additionally, Silvesglow et al. also concluded that to promote healthcare safety, it is crucial for professionals to have the skills to work in a team and establish open communication [35].

Obstacles to quality and safety.

Concerning the barriers to achieving quality and safety in home care, we have the lack of time; the lack of a protocol to assess quality, and the absence of a formal instrument that can assess the care provided by the patient and the family caregiver [29]. These insights prompt a deeper reflection on what strategies to implement to counteract the effects of these obstacles, as they may jeopardize the intervention of professionals in the field. As we could verify through these findings, the evaluation instruments are seen as essential pillars for promoting quality and safety, and of course, their absence is an obstacle. Thus, when we are involved in the creation of models of care at home, we know that it is essential to build instruments that allow monitoring and evaluating care, professionals and context. 

It is essential to identify areas and criteria of quality and safety to guarantee good care to the person who needs care at home, and that they feel cared for and satisfied because the care is close and centered on themselves. This systematic review corresponds to the first phase of a research work that we intend to develop, from which we intend to identify relevant scientific evidence that can support and guide the remaining research.

In the second phase of the work, a diagnosis of the situation in Mainland Portugal will be carried out through the application of a questionnaire that intends to identify which quality and safety methodologies exist in home care in Primary Health Care. In the final phase, we intend to use focus groups made up of Stakeholders from the areas of program planning and coordination, teaching and home care to seek consensus on the areas and criteria of quality and safety. To later be integrated into the construction of a quality and safety model for home care.

The results of the investigation intend to expand knowledge on the subject and be able to contribute to the improvement of quality and safety in home care, and to build a model that allows the provision of evaluation, intervention and management guidelines.

The economic and human resources used for this study corresponded to the platforms provided by the University of Évora and the time made available by the study authors.

## 5. Conclusions

The main aim of this systematic review was to identify areas, dimensions and criteria of quality and safety in home care in the last five years. Thus, three areas, their dimensions and criteria were identified.

(1)Interventions with the patient with the dimension of proximity and patient-centered care, and as operational criteria: individualized care; continuity and organization of care; interpersonal aspects; multidimensional assessment of needs; relationship of proximity and trust; access to the health system.(2)Management of care and services with the dimension of management and clinical governance of care, and as criteria: communication, flow of information and collaboration; enhance the use of clinical data; integrated service pathways; community networks and resources; technology and artificial intelligence; awareness of the purpose of quality measures; s systematize and organize care; evaluation instruments; evaluate expected results; context with adequate conditions; focus attention on the quality of services and health promotion in the community.(3)Training and professional development, with the dimension of skills and training of professionals, and as criteria: improving the background of professionals; care experts; necessary resources.

All these areas, dimensions and criteria are considered structuring pillars for a home care service to guarantee quality and safety for people and communities.

Elements that could hinder quality and safety in home care were also identified, such as the lack of time, the lack of a protocol to assess quality and the absence of formal assessment of care by the patient and family caregiver. Therefore, decision-makers and health professionals must be aware of these aspects so as not to compromise the quality and safety of services and home care. Therefore, it is essential to raise awareness of the aims and measures of quality, as well as the need to systematize and organize care and the importance of creating and using instruments that allow the evaluation of expected results. Similarly, to make it possible to operationalize the assessment of the quality of services and care, to verify whether they are effective, safe, equitable, timely and efficient, and to determine whether they promote increased health literacy in the community, which is extremely important for patients, family and caregivers are empowered to have autonomy in their care.

Although research has revealed important areas, dimensions and criteria of quality and safety related to patient-centered care at home, in the studies found there are gaps related to family members and/or informal caregivers, who are equally providers and recipients of healthcare.

## 6. Limitations of the Current Study

This systematic review of the literature only included studies in English and Portuguese, so language bias may exist. Similarly, only studies from the last 5 years were included, and secondary studies and those that did not present an abstract or full article were also excluded, which may have limited the study.

## Figures and Tables

**Figure 1 ijerph-20-07189-f001:**
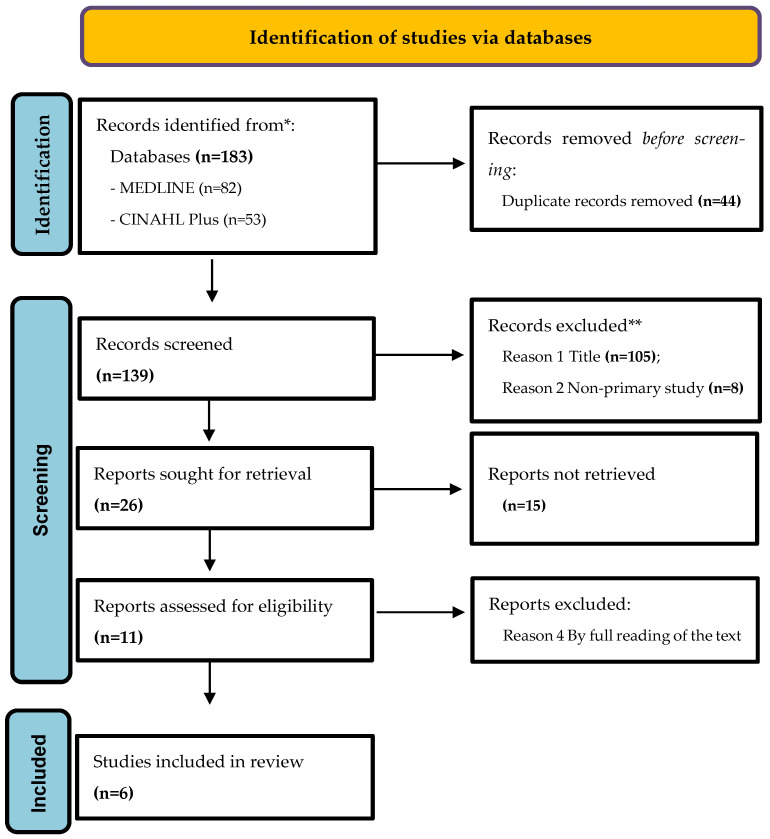
PRISMA 2020 flow diagram. * Consider, if feasible, reporting the number of records identified from each database or register searched (rather than the total number across all databases/registers). ** If automation tools were used, indicate how many records were excluded by a human and how many were excluded by automation tools.

**Table 1 ijerph-20-07189-t001:** Description of included studies (*n* = 6)—Period January 2017 and November 2022.

Studies	Aim	Method	Data Collection Instruments	Participants	Results
Nakrem, S., Kvanneid, K. (2022) [27]	Obtain information on healthcare professionals’ perceptions of the quality of home healthcare and factors that put patient safety at risk	Qualitative	semi-structured interviews	8 Health professionals from home care services	Categories of factors to provide quality services: (1) Competence appropriate to the workplace; (2) Communication, information flow and collaboration; (3) Continuity and organization of care; (4) Resources.
Brunelli, L., et al. (2022) [28]	Develop a validated accreditation tool for home care	Qualitative	consensus technique (relevance and feasibility (RF); agreement)	21 Specialists	-Individualized care project with the best scores (RF 8.4; 100% agreement); (1) Multidimensional needs assessment (RF 8.1, 86% agreement); (2) Access to the integrated health and social assistance system (RF 8.1; 86% agreement); (3) Integrated care pathways (RF 7.5; 81% agreement).
Haex, R., et al. (2020) [29]	Understand the needs of the client, formal/informal caregivers and managers responsible for the policy in measuring the quality of care experienced by the client in home care	Qualitative	Focus groups and semi-structured interviews	4 group interviews and 25 semi-structured interviews with key stakeholders	Two main purposes for measuring the quality of care experienced: (1) Improving the primary care process for individual clients; (2) Learn and improve the home care team. Home care organizations should aim for quality, opting for experienced measures to improve the primary care processes of individual clients. The results also underline the importance of adopting, in addition to quantitative assessments, more narrative assessment methods that support open communication about care experiences.
Kattouw, et al. (2019) [30]	Increase knowledge about how municipal decision-makers justify their choices, describe and emphasize quality, patient safety and health promotion in the organization of community nursing service	Qualitative	Interviews in focus groups and individual interviews/Comparative case study of two municipalities	Councilors from each of the two municipalities and heads of municipal health services	-The formal recommendations made by the municipal administration have a significant impact on how the community nursing service is organized in each municipality. -Evidence-based knowledge and health promotion are of limited importance. -Concerns about quality and safety have a moderate impact in one of the municipalities, while in the other they have little impact. -There seems to be a considerable gap between the decision-maker and patient levels.
Elliot, J., et al. (2020) [31]	Determining how providers and administrators use mandatory clinical data captured in the Home Care Reporting System (HCRS) through interRAI-HC tools, there is an opportunity to improve their use	Qualitative	Individual interviews and focus group interviews	11 coordinators of home care organizations	Participants recognized challenges in using this data, such as: -Leveraging interRAI data in the complex home care environment is limited by several factors: the general “newness” of this data in many jurisdictions; the large volume of data; limited capacity and resources to interpret and analyze the data; and connectivity issues in rural connectivity areas. Participants recognized and appreciated the training received and made several recommendations for further training. Clinical datasets have the potential to improve quality and inform decision-making. However, to use this data: (1) Home care agencies need additional training, staffing and support; (2) Additional training and resources for organizations to improve the use of available data and outcomes for individuals receiving home care services.
Malley, J., et al. (2019) [32]	Understand the relationship between care experience and quality of life in long-term home care - Measure perceptions of the care process - Measure the patient’s quality of life	Cross-sectional study	A standardized questionnaire, including questions on QoL-outcomes derived from the ASCOT measure	14,172 people aged 65 and over using home care services	Interpersonal aspects of care: (1) The team’s responsiveness and dedicated behavior have a greater relationship with ASCOT than those related to the organization of care by the professional: such as punctuality and continuity of care; (2) There is an increase of ten percentage points in the former associated on average with an increase of 1.9 percentage points in the ASCOT and an increase of ten percentage points in the latter associated on average with an increase of 0.3 percentage points in the ASCOT; (3) Perceptions of the care experience, particularly those related to aspects of interpersonal care, have an important association with quality of life outcomes.

**Table 2 ijerph-20-07189-t002:** Results: Quality and safety areas, dimensions and criteria.

Research questions: ✓ What is the state of the art in the areas of quality and safety in home care? ✓ What areas and criteria must exist for the construction of a health care model that guarantees quality and safety for patients at home?	Studies
1st Area: Intervention with the Patient	
Dimension: Proximity and Patient-Centered Care	
Criterion: Individual Care Plan - Individualized service [28]; - Continuity and organization of care [27]; - Interpersonal aspects [32]; - Multidimensional needs assessment [28,30].	[27,28,30,32]
Criterion: Proximity of Professionals to Patient and family - Relationship of proximity and trust [30]; - Access to the health system [28].	[28,30]
2nd Area—Management of Care and Services	
Dimension: Management of care and Clinical Governance of Services	
Criterion: Integrated Health Care Model - Communication, information flow and collaboration [27]; - To enhance the use of clinical data [31]; - Integrated service routes [28]; - Networks and community resources [30]; - Technology and Artificial Intelligence [31].	[27,28,30,31]
Criterion: Management of Goals - Awareness of the purpose of quality measures [29]; - Systematize and organize care [30]; - Assessment instruments [31]; - Evaluate the expected results [30].	[29,30,31]
Criterion: Context Management - Context with adequate conditions [27]; - Focus attention on the quality of services and health promotion in the community [30].	[27,30]
3rd Area—Training and Professional Development	
Dimension: Skills and training of professionals	
- Improve the background of professionals [29]; - Experts in care [29]; - Resources needed [27].	[27,29]

**Table 3 ijerph-20-07189-t003:** Obstacles to the quality and safety process.

Elements of Hinder Quality and Safety	Studies
○ Shortage of time [29]; ○ Lack of a protocol to assess quality [29]; ○ Absence of formal assessment of care for the patient and the family caregiver [29].	[29]

## Data Availability

Not applicable.

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
