# Peer review of "Quality and Safety of Proximity Care Centered on the Person and Their Domiciliation: Systematic Literature Review"

_ijerph, 2023, doi:10.3390/ijerph20247189_

Round 1

Reviewer 1 Report

Comments and Suggestions for Authors

The article is well-prepared for publication, with a clear structure, comprehensive methodology, and relevant findings. The topic is timely and significant, and the study adds valuable insights to the existing literature on home care quality and safety. Nonetheless, minor improvements in justifying certain methodological choices and expanding on the implications of the findings would enhance its readiness for publication. 

Specific Comments

-         The introduction could benefit from a more explicit statement of the research gaps the review aims to address.

-         The decision to limit the search to articles published in English and Portuguese should be justified, as it could exclude relevant studies in other languages.

-         The search strategy, including databases and search terms, is adequately detailed, but the rationale for selecting these specific databases would enhance the methodological transparency.

-         The systematic approach to study selection and data extraction is commendable. However, the process of resolving discrepancies, if any, during study selection and data extraction could be better detailed.

-         The use of the Joanna Bring Institute (JBI) tools for quality assessment is appropriate, but a brief explanation of why these particular tools were chosen would be beneficial.

-         The narrative approach to data synthesis is suitable given the diverse methodologies of the included studies. However, a brief discussion on how this approach helps in addressing the research questions would be informative.

Comments on the Quality of English Language

The writing is clear, concise, and well-organized, facilitating easy comprehension of the complex subject matter. The abstract provides a succinct overview of the study's purpose, methods, and key findings, which is helpful for readers.

Author Response

Thank you very much for your timely comments and suggestions, which we seek to integrate to improve the article and which will also serve as a lesson for future improvements.

Specific Comments

- The introduction could benefit from a more explicit statement of the research gaps the review aims to address.

Response: Integrated into the text – “Also, in the review carried out by Al Anazi et al. (2020), shows a clear gap in the literature on the quality of home care in Arab countries, emphasizing the need for more studies, es-pecially quality studies on care in home health environments [16].”

- The decision to limit the search to articles published in English and Portuguese should be justified, as it could exclude relevant studies in other languages.

Response: Integrated into the text – “written in Portuguese and English, as we were looking for a similarity in care systems, and it seemed to us that they would be covered in articles written in Portuguese and English”

- The search strategy, including databases and search terms, is adequately detailed, but the rationale for selecting these specific databases would enhance the methodological transparency.

Response: This time we accessed through ESCOhost Resarch Databases: CINAHL Plus with Full Text, MEDLINE with Full Text and Coleção Psicologia e Ciências do Comportamento, but we will try to also access other platforms in the future.

- The systematic approach to study selection and data extraction is commendable. However, the process of resolving discrepancies, if any, during study selection and data extraction could be better detailed.

Response: Integrated into the text – “The studies resulting from the search in each database were exported to Mendeley and duplicates removed using an electronic methodology in addition to the manual activity of two reviewers by reading the articles.”

And

“Subsequently, the full text evaluation phase was carried out using the same method, in order to minimize biases, it was also not necessary to use a third reviewer.”

- The use of the Joanna Bring Institute (JBI) tools for quality assessment is appropriate, but a brief explanation of why these particular tools were chosen would be beneficial.

Response: Integrated into the text – “The choice of these specific tools was due to the different methodologies of the studies in-cluded, for allowing us to evaluate the methodological quality of each study and deter-mine the extent to which the study addressed the possibility of bias in its design, conduct and analysis. And JBI's approach considers the best available evidence, the context in which care is delivered, the individual patient and the professional judgement and exper-tise of the health professional.”

- The narrative approach to data synthesis is suitable given the diverse methodologies of the included studies. However, a brief discussion on how this approach helps in addressing the research questions would be informative.

Response: Integrated into the text – “The studies had different methodologies, the synthesis of data and analysis of the results was of a narrative nature, structured to answer the defined questions, allowing an individual analysis of the content and/or structure of the studies until synthesizing a metanarrative. A quantitative analysis was not feasible [26].

We also revisited a small edition of the English language, improved and updated the introduction and cited references, as well as the discussion and conclusion of the study.

Thank you very much for your timely comments and suggestions, which we sought to integrate into the article and also served as learning for improvement in the future.

Very recognized.

Reviewer 2 Report

Comments and Suggestions for Authors

The topic of the study is very interesting and certainly represents an emerging global need.However, my question is whether the results are sufficient for a comprehensive assessment of the state of the art.

In this context, I would like to ask why Pubmed, which is one of the leading biomedical databases, has not been included as a source of data.

Furthermore, while the conclusions refer to a five-year study, it is actually a six-year study, albeit an incomplete one, as it covers the period from January 2017 to November 2022 (Why wasn't December 2022 included?).

Introduction

The introduction should be tightened up a bit, in my opinion.For example, one could think of eliminating references to Portuguese documents, which, presented in this way, do not seem to me to bring fundamental concepts to the objective of the study.In addition, it could be considered not to include in the introduction the results on the 3 areas identified, which are then dealt with in detail in the discussion. The introduction should not pre-empt discussion and conclusions.

pag. 2 

line 75 - Perhaps the date in brackets could be removed, as the reference is explicitly there.

line 94 - Is it possible to insert a bibliographical reference related to the Donabedian model?

Materials and Methods

Open the acronym : International Prospective Register of Systematic Reviews (PROSPERO) 

pag. 5 line 205-207

It is proposed to delete the following sentence as a repetition (already mentioned a few lines earlier)

"The strategy was adapted according to each database 205

and limited to the time period between January 2017 and November 2022. Studies pub- 206

lished in Portuguese and English were considered".

Results

As there are only 6 articles identified, in addition to 3 areas emerged, some more data could be provided on these studies, which target group they studied, which clinical conditions.It would be helpful if there was a better explanation of how the selected studies contributed to answering the question or led to a progression of 'cognitive' development on the topic. 

It also seems to me that there is no mention of economic and human resources in the discussion.

Conclusion

The conclusions call for a critical assessment of the extent and reliability of the information on the subject, so it would be helpful not to repeat the findings but to provide a more discursive summary of the current state of knowledge on the subject, clarifying whether the research has been adequate and whether there are gaps or missing areas.

References 

Some references may be a bit old

In general, I would revise the whole article. I would remove some repetitions and improve the not always fluent English.

Comments on the Quality of English Language

In general, I would revise the whole article. I would remove some repetitions and improve the not always fluent English.

Author Response

Thank you very much for your timely comments and suggestions, which we seek to integrate to improve the article and which will also serve as a lesson for future improvements.

Comments and Suggestions for Authors

The topic of the study is very interesting and certainly represents an emerging global need.However, my question is whether the results are sufficient for a comprehensive assessment of the state of the art.

Response: It is the first part of an investigation that we are carrying out, to serve as a basis for its development. For what we also integrated in the article that "This systematic review corresponds to a first phase of research work that we intend to develop, from here we intend to identify relevant scientific evidence that can support and guide the remaining research. In a second phase of the work, a diagnosis of the situation in Mainland Portugal will be carried out through the application of a questionnaire that aims to identify which quality and safety methodologies exist in home care in Primary Health Care. In a final phase, we intend to use focus groups made up of Stakeholders from the areas of planning and coordination of programs, teaching and home care to seek consensus on the areas and criteria of quality and safety. To later be integrated into the construction of the quality and safety model in home care. The results of the investigation aim to expand knowledge on the subject and contribute to improving the quality and safety of home care, and also to build a model that allows the provision of assessment, intervention and management guidelines.”

In this context, I would like to ask why Pubmed, which is one of the leading biomedical databases, has not been included as a source of data.

Response: This time we accessed through the ESCOhost Resarch databases: CINAHL Plus with Full Text, MEDLINE with Full Text and the Psychology and Behavioral Sciences Collection. But we really appreciate the suggestion and next time we will also try to access other platforms, namely Pubmed.

Furthermore, while the conclusions refer to a five-year study, it is actually a six-year study, albeit an incomplete one, as it covers the period from January 2017 to November 2022 (Why wasn't December 2022 included?).

Response: Due to academic timing of the main researcher's PhD.

  • Introduction

The introduction should be tightened up a bit, in my opinion.For example, one could think of eliminating references to Portuguese documents, which, presented in this way, do not seem to me to bring fundamental concepts to the objective of the study.In addition, it could be considered not to include in the introduction the results on the 3 areas identified, which are then dealt with in detail in the discussion. The introduction should not pre-empt discussion and conclusions.

Response: Content and references to Portuguese documents were eliminated.

line 75 - Perhaps the date in brackets could be removed, as the reference is explicitly there.

Response: Removed date in brackets.

line 94 - Is it possible to insert a bibliographical reference related to the Donabedian model?

Response: The bibliographic reference related to the Donabedian model was inserted.

  • Materials and Methods

Open the acronym : International Prospective Register of Systematic Reviews (PROSPERO) 

  • 5 line 205-207

It is proposed to delete the following sentence as a repetition (already mentioned a few lines earlier)

"The strategy was adapted according to each database and limited to the time period between January 2017 and November 2022. Studies published in Portuguese and English were considered".

Response: The phrase was deleted.

  • Results

As there are only 6 articles identified, in addition to 3 areas emerged, some more data could be provided on these studies, which target group they studied, which clinical conditions.It would be helpful if there was a better explanation of how the selected studies contributed to answering the question or led to a progression of 'cognitive' development on the topic. 

It also seems to me that there is no mention of economic and human resources in the discussion.

Integrated into the text – “As it is essential to identify areas and criteria of quality and safety to guarantee good care to the person who needs care at home, and also that they feel cared for and satisfied because the care is close and centered on themselves. This systematic review corresponds to a first phase of a research work that we intend to develop, from here we intend to iden-tify relevant scientific evidence that can support and guide the remaining research.

In a second phase of the work, a diagnosis of the situation in Mainland Portugal will be carried out through the application of a questionnaire that intends to identify which quality and safety methodologies exist in home care in Primary Health Care. In a final phase, we intend to use focus groups made up of Stakeholders from the areas of program planning and coordination, teaching and home care to seek consensus on the areas and criteria of quality and safety. To later be integrated into the construction of the quality and safety model for home care.

The results of the investigation intend to expand knowledge on the subject and be able to contribute to the improvement of quality and safety in home care, and also to build a model that allows the provision of evaluation, intervention and management guidelines.

With regard to the economic and human resources used for this study, they corre-sponded to the platforms provided by the University of Évora and the time made available by the study authors.”

  • Conclusion

The conclusions call for a critical assessment of the extent and reliability of the information on the subject, so it would be helpful not to repeat the findings but to provide a more discursive summary of the current state of knowledge on the subject, clarifying whether the research has been adequate and whether there are gaps or missing areas.

Integrated into the text – “Although research has revealed important areas, dimensions and criteria of quality and safety related to patient-centered care at home, in the studies found there are gaps re-lated to family members and/or informal caregivers, who are equally providers and recip-ients of healthcare.”

  • References 

Some references may be a bit old

Response: Integrated updated references in the text and bibliography.

In general, I would revise the whole article. I would remove some repetitions and improve the not always fluent English.

We also revisited a small edition of the English language, removing some repetitions, improving and updating the introduction and cited references, as well as the research design, discussion and conclusion of the study, with a great contribution from your comments and suggestions.

Thank you very much for your timely comments and suggestions, which we sought to integrate into the article and also served as learning for improvement in the future.

Very recognized.

Reviewer 3 Report

Comments and Suggestions for Authors

Thank you for your paper titled, “Quality and safety of proximity care centered on the person and 2 their domiciliation: Systematic Literature Review”. Overall, this study was conducted properly by registering on PROSPERO and following the PRISMA guideline. 

The main concern pertains to the literature search that is not updated. In fact, it was finished on November 1, 2022, quite a year ago. I strongly encourage authors to implement their search and potential findings to date.

Some other minor suggestions for the authors to consider:

  1. What is the reason to restrict the search to 2017? Why haven’t the authors considered studies before 2017?

  2. The results of this study may be limited by narrative synthesis. Given the fact that only 6 study was included and the study designs were heterogeneous across the included studies, the quantitative analysis was not feasible.

  3. To screen duplicates, did the authors use an electronic methodology beyond hand activity?

Author Response

Thank you very much for your timely comments and suggestions, which we seek to integrate to improve the article and which will also serve as a lesson for future improvements.

Comments and Suggestions for Authors

The main concern pertains to the literature search that is not updated. In fact, it was finished on November 1, 2022, quite a year ago. I strongly encourage authors to implement their search and potential findings to date.

Response: New research and discoveries have been integrated to date. We also improved and updated the introduction and cited references, as well as the discussion and conclusion of the study.

For example, it was integrated into the texto:

- “For these reasons, home care currently represents a progressive investment area, where healthcare services and the scientific community express concern and interest in developing knowledge that enables the creation of circuits and tools to ensure the quality of care [4, 5]. Additionally, there is a focus on building strategies to prevent adverse events in home care [6].”

- “Certainly, still regarding the integrated home healthcare model, for it to function ad-equately and effectively, it is imperative that there is communication among professionals, patients, family/caregivers [27], and that information flows among the different stake-holders, as well as everyone collaborating in feeding the model around a common goal, which is the well-being of the patient. Additionally, the potential use of patient clinical data can limit lapses or failures in care regarding the multiple needs of the patient and fa-cilitate the organization and management of the necessary human and material resources. In a study conducted by Silverglow et al (2023) in Sweden, it became evident that the scar-city of information about the user between healthcare units and home care can constitute a barrier to the safety of healthcare [35].”

- “On the other hand, communication between the different players is also important, as emphasised of the study Silvesglow et al (2023) demonstrated that communication prob-lems, both between care professionals and with care unit managers, can weaken patient safety in home care [35].”

- “On the other hand, the use of technology can contribute to solving some problems in terms of human resource shortages and reducing errors, as mentioned Kajander-Unkuri et al. (2023), using a robot for medication management had a notable effect on decreasing the use of working time of home care professionals [36].”

- “In addition to these aspects, Silvesglow et al (2023) also concluded that to promote healthcare safety, it is crucial for professionals to have skills to work in a team and estab-lish open communication [35].”

Some other minor suggestions for the authors to consider:

  1. What is the reason to restrict the search to 2017? Why haven’t the authors considered studies before 2017?

Response: The research intended to obtain the design of recent years in relation to the quality and safety of patients at home, and did not aim to explore the evolution of home care. However, it seems to us to be a very timely suggestion for future studies in this field.

  1. The results of this study may be limited by narrative synthesis. Given the fact that only 6 study was included and the study designs were heterogeneous across the included studies, the quantitative analysis was not feasible.

Response: It was not possible given the studies found, but in future studies we will take this into account.

Integrated into the text – “A quantitative analysis was not feasible.”

  1. To screen duplicates, did the authors use an electronic methodology beyond hand activity?

Response:  Yes, and integrated into the text – “The studies resulting from the search in each database were exported to Mendeley and duplicates removed using an electronic method-ology in addition to the manual activity of two reviewers by reading the articles.”

Thank you very much for your timely comments and suggestions, which we sought to integrate into the article and also served as learning for improvement in the future.

Very recognized.